# FROM ZERO TO TURBULENCE: GENERATIVE MODELING FOR 3D FLOW SIMULATION

**Marten Lienen, David Lüdke, Jan Hansen-Palmus, Stephan Günnemann**
Department of Informatics & Munich Data Science Institute
Technical University of Munich, Germany
{m.lienen,d.luedke,j.hansen-palmus,s.guennemann}@tum.de

## ABSTRACT

Simulations of turbulent flows in 3D are one of the most expensive simulations in computational fluid dynamics (CFD). Many works have been written on surrogate models to replace numerical solvers for fluid flows with faster, learned, autoregressive models. However, the intricacies of turbulence in three dimensions necessitate training these models with very small time steps, while generating realistic flow states requires either long roll-outs with many steps and significant error accumulation or starting from a known, realistic flow state—something we aimed to avoid in the first place. Instead, we propose to approach turbulent flow simulation as a generative task directly learning the manifold of all possible turbulent flow states without relying on any initial flow state. For our experiments, we introduce a challenging 3D turbulence dataset of high-resolution flows and detailed vortex structures caused by various objects and derive two novel sample evaluation metrics for turbulent flows. On this dataset, we show that our generative model captures the distribution of turbulent flows caused by unseen objects and generates high-quality, realistic samples amenable for downstream applications without access to any initial state.

## 1 INTRODUCTION

CFD is an integral component of engineering today, and significant computing resources are spent on it every day at scales small and large. Engineers simulate fluid flows to maximize the throughput in chemical plants, optimize the energy yield of wind turbines, or improve the efficiency of aircraft engines (Reguly et al., 2016). The widespread use of these simulations makes their acceleration with machine learning highly impactful (Raissi et al., 2019; Rackauckas et al., 2021; Kochkov et al., 2021; Li et al., 2021).

A CFD simulation consists of a discretized geometry represented as a grid or mesh, boundary conditions that specify, for example, the position and behavior of walls and inlets, and initial conditions that provide a known state of the flow. Since the turbulent flow's behavior is unknown, these initial conditions are usually specified as constants or smooth approximations of the expected flow. To produce realistic turbulent states, a numerical solver solves the Navier-Stokes equation

$$\frac{\partial \boldsymbol{u}}{\partial t} = \nu \nabla^2 \boldsymbol{u} - \frac{1}{\rho} \nabla p - (\boldsymbol{u} \cdot \nabla)\boldsymbol{u} \tag{1}$$

forward in time until the flow transitions from the simplified initial conditions to fully developed turbulence. The equation describes the relationship between velocity $\boldsymbol{u}$ and pressure $p$ for a viscous fluid with kinematic viscosity $\nu$ and constant density $\rho$. For liquids, Eq. (1) is often supplemented with the incompressibility assumption $\nabla \cdot u = 0$, and we do so, too.

While numerical solvers can simulate most kinds of flows to high precision, large simulations can be very slow, especially in three dimensions (3D). A prominent approach to accelerate these simulations is to emulate the numerical solver with autoregressive models parameterized by neural networks. Many works have demonstrated formidable speedups in two dimensions (2D) for scenarios of varying complexity and all kinds of models of this class (Horie & Mitsume, 2022; Pfaff et al., 2021; Shu et al.,

---

Find code and data at https://cs.cit.tum.de/daml/generative-turbulence.

2023; Yang & Sommer, 2023; Janny et al., 2023; Obiols-Sales et al., 2020; Zhao et al., 2022; Kochkov et al., 2021; Wang et al., 2020; Li et al., 2021; Brandstetter et al., 2022) and several benchmark datasets for the 2D Navier-Stokes equation have been published in recent years (Bonnet et al., 2022; Gupta & Brandstetter, 2022; Takamoto et al., 2022; Otness et al., 2021). In comparison, 3D results are much sparser in terms of models (Stachenfeld et al., 2022) as well as datasets (Takamoto et al., 2022; Li et al., 2008), even though 3D simulations offer the highest acceleration potential because of their enormous computational costs.

Autoregressive models usually outperform numerical solvers by taking larger time steps without diverging. However, in 3D, turbulent flows exhibit a stark difference to their 2D counterparts hampering this strategy. Because short-lived, small-scale vortices drive the chaotic turbulence in 3D to a much higher degree, their behavior needs to be modeled, which limits the size of the time step of an autoregressive model. At the same time, the complete replacement of the numerical solver requires the model to evolve the flow from the solver's initial state until a turbulent flow state has fully developed, which can require hundreds of steps and lead to catastrophic error accumulation.

However, in many applications, practitioners use numerical simulations not to explore one specific solution trajectory through time but as a proxy to explore the distribution of possible flow states. For example, in design optimization, an engineer might be interested in the possible size, length, and energy of vortices caused by an object, such as an airplane wing, but not in the exact evolution of one particular vortex. In other applications, the formation of certain patterns, such as jets, or the location of stagnation points on the boundary of an object, where the flow velocity is zero, might be important. We conclude that many use cases can be solved by independent snapshots of possible flow states just as well as with a classical numerical simulation.

In this paper, we take an entirely different direction to autoregressive models and propose a generative approach to turbulent flow simulation by directly learning the manifold of all possible turbulent flow states. This way, we overcome the problem of long roll-outs of autoregressive models and skip the initial transition phase as well as all the intermediate states that a numerical solver has to generate to produce a diverse set of flow states.

Our contributions can be summarized as follows:

- We present a novel 3D turbulence dataset with challenging geometries and systematically investigate the specific challenges of forecasting 3D turbulent flows compared to 2D.
- We propose to approach 3D turbulence simulation as a generative task to overcome the roll-out dilemma of autoregressive models. In doing so, we derive an appropriate generative diffusion model and evaluation metrics.
- We verify experimentally that our model captures the distribution of turbulent flows, generalizes to unseen geometries and generates high-quality, realistic samples amenable for downstream application, ultimately eliminating the reliance on numerical solvers.

## 2 TURBULENCE IN TWO AND THREE DIMENSIONS

For a complete introduction to turbulent flows, we refer the reader to (Pope, 2000) and (Mathieu & Scott, 2000). (Ouellette, 2012) gives a great overview of the differences between 2D and 3D turbulence.

In principle, one can solve the incompressible Navier-Stokes-Eq. (1) in two spatial dimensions just as well as in three. However, when we compare the resulting flow fields of two simulations in Fig. 1 that only differ in that one simulation domain extends into the third dimension, it becomes obvious that some effect must be exclusive to 3D fluid flows. The 3D data exhibits many small-scale features, while the two-dimensional flow field is dominated by relatively smooth and orderly large-scale structures. But why is that?

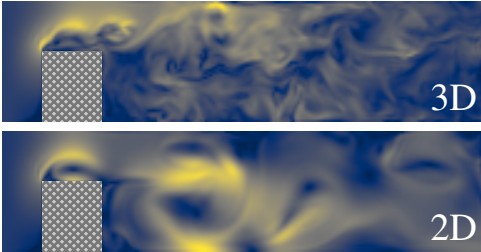

Figure 1: The same simulation exhibits vastly different qualities when the solver operates in three dimensions compared to two.

One of the defining features of turbulence in 3D is the energy cascade, a process that moves kinetic energy from large-scale vortices to ever smaller scales until it is finally directly converted to thermal energy at the molecular scale. The main drivers of this process are two effects called vortex stretching and strain self-amplification (Carbone & Bragg, 2020). Together, they are responsible for the continued creation of vortex structures at all scales and the high-frequency features characteristic of 3D turbulence.

In two dimensions, both of those effects vanish mathematically, ultimately due to the fact that the vorticity $\boldsymbol{\omega} = \nabla \times \boldsymbol{u}$, which describes the local rotationality of the flow, is orthogonal to the velocity gradient in 2D. As a consequence, the energy cascade is only driven by a weaker interaction effect between vorticity and strain, which inverts the energy cascade (Johnson, 2021). The inverted energy cascade, in contrast to 3D, transports energy from the small to the large scales and creates more homogeneous, long-lived structures.

Temporally, the lifetime of vortex structures depends on their size, with smaller structures decaying faster than larger ones (Lozano-Durán & Jiménez, 2014). From a machine learning perspective, this means that *forecasting the future trajectory of a fluid simulation in 3D requires smaller time steps than for 2D data* to resolve the nonlinear behavior of small-scale features under the Navier-Stokes equation.

## 3 AUTOREGRESSIVE FORECASTING IN 3D

Autoregressive models are popular surrogate models for numerical solvers of time-dependent partial differential equations (PDEs) and, in particular, the Navier-Stokes equation, as they promise faster runtimes than full simulations. They emulate numerical solvers by learning to predict the next state from the current state and, optionally, a set of previous states, which enables approximating fluid dynamics over time via an iterative roll-out. While autoregressive models offer advantages, i.e. faster evaluation, ease of parallelization, and the ability to use larger time steps, their possible runtime improvement is directly limited by how many solver steps the model can compress into one model evaluation.

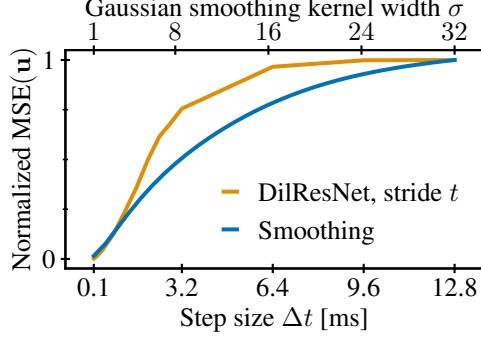

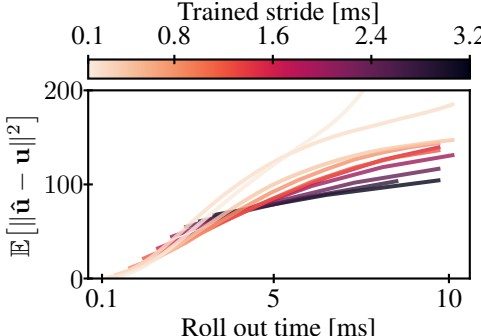

Figure 2: 1-step forecast MSE of $u$ increases at a rate comparable to Gaussian smoothing of states to remove small-scale features.

Figure 3: Error accumulates quickly for longer roll-outs even when the time step is sufficiently small.

As described in Section 2, 3D turbulent flows contain many small-scale features that evolve nonlinearly on short timescales. In our experiment shown in Fig. 2, we have trained a Dilated ResNet (DilResNet) on one-step-ahead prediction with varying step size $\Delta t$. While the model achieves a very small error for the smallest time step, the one-step forecasting error grows as we increase the time step that we train the model for. Notably, the error curve aligns closely with the curve resulting from smoothing of the target with a Gaussian kernel of increasing width. By smoothing the velocity field with a Gaussian kernel of variance $\sigma^2$, we remove features of scale approximately $\sigma$ and below. The alignment of the curve then indicates that the model trained to predict $\Delta t$ ahead fails to predict features below a cut-off scale $\sigma$ that scales linearly with $\Delta t$. In fact, visual inspection of the forecasts reveals that the model makes increasingly smooth predictions as the model's time step increases. Conversely, this means that the model needs to be trained and evaluated with a very small time step to even have a chance to forecast fully resolved future states of 3D turbulent flows over multiple steps.

Ultimately, modeling 3D turbulent flow simulations poses an inescapable dilemma for autoregressive models. On the one hand, turbulent flows exhibit significant small-scale behavior that is important to the overall behavior of the flow because of backscattering effects from small to large scales, which requires models to be trained with a small step size. On the other hand, these simulations need to be forecast for hundreds of steps into the future to traverse the transition phase from an initial simulation state to a fully developed turbulent flow, which necessitates long roll-outs. However, autoregressive models trained with smaller step sizes and, consequently, more roll-out steps suffer from more error accumulation in their forecasts, cf. Fig. 3. Conversely, models trained with large time steps accumulate less error but start with large amounts of error from the first step.

## 4    GENERATIVE MODELING FOR 3D TURBULENT FLOWS

We have seen that forecasting 3D turbulent flows with their chaotic nature and many small-scale, short-lived features is challenging, in particular for autoregressive models. Given their chaotic nature and sensitivity to infinitesimal perturbations, turbulent flows can be understood as stochastic processes (Pope, 2000), i.e. the solution trajectory $\{(\boldsymbol{u}, p)_t\}_{t \in \mathbb{R}^+}$ given some initial conditions follows the distribution $\mathrm{p}(\{(\boldsymbol{u}, p)_t\}_{t \in \mathbb{R}^+} \mid \boldsymbol{u}_0, p_0)$. Yet, many applications do not require an exact trajectory but rather a selection of turbulent flow states that represent the set of all possible flows well. Thus, we propose to circumvent the roll-out dilemma of autoregressive models by modeling the marginal distribution $\mathbb{E}_t[\mathrm{p}((\boldsymbol{u}, p)_t \mid \boldsymbol{u}_0, p_0, t)] = \mathrm{p}((\boldsymbol{u}, p) \mid \boldsymbol{u}_0, p_0)$ directly, capturing the distribution of turbulent flows with a generative model.

### 4.1    GENERATIVE TURBULENCE SIMULATION

Consider a simulation domain $\Omega \subset \mathbb{R}^3$ with boundary conditions $\mathcal{B}$ and initial conditions $\boldsymbol{u}^{(0)} : \Omega \to \mathbb{R}^3$, $p^{(0)} : \Omega \to \mathbb{R}$ for velocity and pressure, where $\mathcal{B}$ is chosen such that the flow becomes turbulent. The fields $\boldsymbol{u}_{\mathrm{sol}} : \mathbb{R}^+ \times \Omega \to \mathbb{R}^3$ and $p_{\mathrm{sol}} : \mathbb{R}^+ \times \Omega \to \mathbb{R}$ solving the Navier-Stokes-Eq. (1) describe the resulting, turbulent flow. In the following, we define $\boldsymbol{X} = (\boldsymbol{u} \parallel p)$ to denote the concatenation of velocity and pressure. Now we define the probability distribution over all attained flow states after a time $t$,

$$\mathrm{p}_{\geq t}(\boldsymbol{X} \mid \boldsymbol{X}^{(0)}, \mathcal{B}) = \lim_{T \to \infty} \mathbb{E}_{t' \in [t, T]} \left[ \delta_{\boldsymbol{X}_{\mathrm{sol}}(t')} \right]. \tag{2}$$

If the initial state $\boldsymbol{X}^{(0)}$ is not turbulent, there is a transition phase of length $t_{\mathrm{turb}}$ depending on the domain geometry and $\boldsymbol{X}^{(0)}$ until turbulence is fully developed. Beyond this point, the distribution becomes, remarkably, independent of the initial state, i.e. $\mathrm{p}_{\geq t_{\mathrm{turb}}}(\boldsymbol{X} \mid \boldsymbol{X}^{(0)}, \mathcal{B}) = \mathrm{p}_{\geq t_{\mathrm{turb}}}(\boldsymbol{X} \mid \mathcal{B})$, because of the often assumed ergodicity of turbulence (Galanti & Tsinober, 2004), which says that a turbulent flow will visit all possible flow states. Based on this insight, we define the task of *generative turbulence simulation* as finding a model $\mathrm{p}_{\boldsymbol{\theta}}$ such that

$$\mathrm{p}_{\boldsymbol{\theta}}(\boldsymbol{X} \mid \mathcal{B}) \approx \mathrm{p}_{\geq t_{\mathrm{turb}}}(\boldsymbol{X} \mid \mathcal{B}). \tag{3}$$

Importantly, such a model does not have access to an initial turbulent state $\boldsymbol{X} \sim \mathrm{p}_{\geq t_{\mathrm{turb}}}(\boldsymbol{X} \mid \mathcal{B})$.

### 4.2    DISCRETIZATION

We discretize the simulation domain $\Omega$ into a regular grid of cells $\Omega_h = [W] \times [H] \times [D]$ where $[N]$ denotes the set of integers 1 through $N$. In the following, we will denote cell indices $(i, j, k) \in \Omega_h$ as $\mathbf{i}$. Each cell has one of $K$ types, $\boldsymbol{T_i} \in [K]$, marking a cell as, for example, wall or inlet, which discretizes the location and type of boundary conditions $\mathcal{B}$. To discretize solution fields $\boldsymbol{u}_{\mathrm{sol}}(t)$ and $p_{\mathrm{sol}}(t)$ at time $t$, we define the tensor $\boldsymbol{X_i} = (\boldsymbol{u}_{\mathrm{sol}}(t, \boldsymbol{x_i}) \parallel p_{\mathrm{sol}}(t, \boldsymbol{x_i})) \in \mathbb{R}^4$ where $\boldsymbol{x_i}$ is location of the center of cell $\mathbf{i}$.

See Appendix B for more details on how we construct $\boldsymbol{X}$ and $\boldsymbol{T}$.

### 4.3    GENERATIVE MODEL

We base our model *TurbDiff* on denoising diffusion probabilistic models (DDPMs) (Ho et al., 2020; Sohl-Dickstein et al., 2015), which have been adapted to other domains (Kollovieh et al., 2023; Lüdke et al., 2023). DDPMs are latent-variable models that learn a generative model $\mathrm{p}_{\boldsymbol{\theta}}$ to reverse a fixed

Markov chain $q(\boldsymbol{x}_n|\boldsymbol{x}_{n-1})$, which gradually transforms a data sample $\boldsymbol{x}_0$ over $N$ steps into Gaussian noise until no information remains, i.e. $q(\boldsymbol{x}_N \mid \boldsymbol{x}_0) \approx \mathcal{N}(\mathbf{0}, \boldsymbol{I})$. Training occurs by minimizing the KL-divergence between $p_{\boldsymbol{\theta}}(\boldsymbol{x}_{n-1} \mid \boldsymbol{x}_n)$ and $q(\boldsymbol{x}_{n-1}|\boldsymbol{x}_0, \boldsymbol{x}_n)$, so that $p_{\boldsymbol{\theta}}$ approximates the true posterior. The reverse process then constitutes a powerful sampling mechanism that transforms sampled noise $\boldsymbol{x}_N \sim \mathcal{N}(\mathbf{0}, \boldsymbol{I})$ into a data sample by iteratively sampling from $p_{\boldsymbol{\theta}}(\boldsymbol{x}_{n-1} \mid \boldsymbol{x}_n)$ and, at the same time, formally defines our generative model as $p_{\boldsymbol{\theta}}(\boldsymbol{x}_0) = \int p(\boldsymbol{x}_N) \prod_{n=1}^{N} p_{\boldsymbol{\theta}}(\boldsymbol{x}_{n-1} \mid \boldsymbol{x}_n) \mathrm{d}\boldsymbol{x}_1 \ldots \boldsymbol{x}_N$.

We adapt DDPM and include Dirichlet boundary conditions to guide the diffusion process similar to the inpainting method proposed by (Lugmayr et al., 2022). Dirichlet conditions fix the value of velocity or pressure at a specific location to a given value, e.g. $\boldsymbol{u} = 0$ at the walls or the given inflow velocity at the inlet. Therefore, we sample the values at these locations from fixed distributions instead of the learned distribution $p_{\boldsymbol{\theta}}$. This ensures that the values at the boundaries converge towards their prescribed values throughout the sampling process, guiding the sample as a whole towards the data distribution.

In particular, we begin by defining a mask $\mathcal{M}_{\mathbf{i}} = \mathbb{I}_{\boldsymbol{T}_{\mathbf{i}}=\text{free}}$ that selects all cells free cells, i.e. those that have no boundary conditions or any other kind of special role. Let $\boldsymbol{X}_0 \sim p_{\geq t_{\text{turb}}}(\boldsymbol{X} \mid \mathcal{B})$ and we want to sample from its distribution by transforming a sample $\boldsymbol{X}_N \sim p(\mathbf{0}, \boldsymbol{I})$. Then, we define our model as

$$p(\boldsymbol{X}_{n-1} \mid \boldsymbol{X}_n, \boldsymbol{T})_{\mathbf{i}} \sim \begin{cases} p_{\boldsymbol{\theta}}(\boldsymbol{X}_{n-1} \mid \boldsymbol{X}_n, \boldsymbol{T})_{\mathbf{i}} & \text{if } \mathcal{M}_{\mathbf{i}} = 1, \\ q(\boldsymbol{X}_{n-1} \mid \boldsymbol{X}_0, \boldsymbol{X}_n)_{\mathbf{i}} & \text{otherwise.} \end{cases} \quad (4)$$

In effect, we denoise the input only in cells that contain actual velocity and pressure data. For other cells, we sample their values from the true posterior $q(\boldsymbol{X}_{n-1} \mid \boldsymbol{X}_0, \boldsymbol{X}_n)$, which is possible because for these cells $\boldsymbol{X}_{0,\mathbf{i}}$ is predetermined from the boundary conditions and therefore always available. Since we also use the same case distinction during training, our model is only trained on actual data cells and not on boundary values.

### 4.4 Parametrizing Posterior and Diffusion Process

Since we discretize the data domain as a three-dimensional, regular grid, we base our model architecture on a U-Net (Ronneberger et al., 2015) extended to 3D (Çiçek et al., 2016). After 4 levels of downsampling, we apply a transformer (Vaswani et al., 2017) with pre-group normalization (Xiong et al., 2020; Wu & He, 2020) to ensure global communication within the model across the whole simulation domain. Within the transformer, we use flash attention to reduce the memory requirements of the transformer from $O(n^2)$ to $O(n)$ (Dao et al., 2022).

For the forward diffusion process, we choose $N = 500$ steps and a noise schedule that scales the noise so that the log-signal-to-noise ratio of the noise grows linearly (Kingma et al., 2021). This schedule ensures that several diffusion steps are spent in the high-signal regime to let the model denoise small details, but at the same time, reaches large enough noise levels to ensure that $q(\boldsymbol{x}_N \mid \boldsymbol{x}_0) \approx \mathcal{N}(\mathbf{0}, \boldsymbol{I})$ despite the long tails of velocity and pressure values in turbulent flows.

For more details on the components and hyperparameters, see Appendix D.

## 5 Related Work

The works of (Drygala et al., 2022) and (Drygala et al., 2023) are most closely related to our research. They prove that generative adversarial networks (GANs) sample from the correct distribution for ergodic systems such as turbulent flows (Galanti & Tsinober, 2004) and train a GAN to generate 2D slices of high-resolution 3D turbulent flow data of the velocity field orthogonal to the slice. Compared to their work, we allow arbitrary geometries by encoding all types of boundary conditions via learned cell type embeddings and we propose a metric that measures the quality of samples as a whole, whereas Drygala et al. examine particular sections of their samples individually.

Several other works have been published on machine learning for 3D fluid flows. (Xie et al., 2017) super-resolve low-resolution turbulent flows with a GAN in a temporally coherent manner. (Matsuo et al., 2021) learn to reconstruct 3D flows from 2D slices for channel flows around a cylinder. (Stachenfeld et al., 2022) propose a model to forecast non-forced, decaying turbulence at a coarser resolution to achieve speedups over numerical solvers. (Kim et al., 2019) predict velocity fields one step ahead for 3D flows in computer graphics applications. They report that they need to train with a

small time step for their latent-space autoregressive model to track small details and generate accurate roll-outs. It is important to note that these works refine, expand, or evolve turbulent flow fields given as input to the model. In contrast, we sample turbulent flows from scratch based solely on the domain geometry and boundary conditions.

## 6 Experiments

### 6.1 Dataset

To evaluate the viability of generative modeling for 3D turbulent flows, we have generated a challenging new dataset. The dataset consists of 45 simulations of an incompressible flow through a $0.4\,\text{m} \times 0.1\,\text{m} \times 0.1\,\text{m}$ channel at $20\,\text{m}\,\text{s}^{-1}$, which results in a Reynold's number of $\text{Re} = 2 \times 10^5$, well into the turbulent regime. The channel is discretized into $192 \times 48 \times 48$ regular grid cells, which balances the spatial resolution to resolve small-scale behavior adequately with memory and storage requirements.

For each simulation, we place a distinct object into the flow, each of which causes a characteristic flow pattern. Fig. 4 shows a subset of the objects and how the object's shape influences the flow. We split the simulations into 27 for training, 9 for validation, and 9 for testing to verify that models can generalize to unseen objects in the flow.

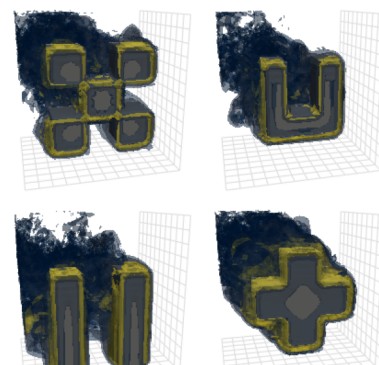

Figure 4: A subset of the objects in our turbulent flow dataset with iso-surfaces of the vorticity magnitude.

We run all simulations for $0.5\,\text{s}$ of physical time with Open-FOAM (Weller et al., 1998), an industrial-grade CFD solver, in large eddy simulation (LES) mode and save a flow state every $0.1\,\text{ms}$. The resulting dataset contains 5000 flow states for each simulation and requires $2.2\,\text{TB}$ of storage after preprocessing.

See Appendix A for more details on our datasets and data generation pipeline.

### 6.2 Baselines

In our experiments, we compare our generative model against two autoregressive models specifically developed for turbulent flows, Turbulent Flow Net (TF-Net) and DilResNet. TF-Net takes inspiration from hybrid RANS-LES modeling of turbulent flows and learns a spatial and temporal filtering with an architecture of multiple U-Nets (Wang et al., 2020). DilResNet is a general-purpose architecture that combines stacks of dilated convolutions with residual connections into an expressive convolutional model with a large receptive field (Stachenfeld et al., 2022). Both models take a set of recent observations $\boldsymbol{X}^{(t-(h-1):t)}$ and map it to a prediction of the system state $\boldsymbol{X}^{(t+1)}$ one step ahead.

For both forecasting models, we compare two different ways to turn them into generative models. For the first, we take a sample $\boldsymbol{X}^{(t)} \sim \text{p}_{\geq t_{\text{turb}}}(\boldsymbol{X} \mid \boldsymbol{T})$ and then apply the model autoregressively 22 times to generate a new sample $\boldsymbol{X}^{(t+22)}$. We chose 22 steps because that is the distance between two steps in our dataset at which they become approximately uncorrelated. Therefore, we consider the model to have transformed $\boldsymbol{X}^{(t)}$ into a new sample $\boldsymbol{X}^{(t+22)}$. Obviously, the model could not replace the numerical solver in this way since we need the solver to generate the input to the generative model in the first place, but it serves us as a very strong baseline because it already starts with a sample from the distribution that we want to sample from. We denote these models by DilResNet-22 and TF-Net-22, respectively.

For the second way to turn our baselines into generative models, we add a small amount of noise to the initial state of the numerical solver and then roll out the models for 200 steps. We chose 200 steps because that is the minimum number of steps the flow needs to develop turbulence for any simulation in our dataset. The small amount of noise leads to independent samples because turbulent flows are chaotic and therefore sensitive to initial conditions, i.e. even an infinitesimal change in initial conditions will produce a completely different trajectory. Of course, these generative variants need

far more roll-out steps, but it completely forgoes the use of a numerical solver, thereby offering the largest speed-up. We denote these models by DilResNet-init and TF-Net-init, respectively.

We list hyperparameters for these baselines in Appendix C.

## 6.3 METRICS

Comparing the quality of turbulent flow samples is a non-trivial task. Unlike in the one-step-ahead prediction of autoregressive models, we do not have a ground truth state but must derive new metrics to quantify the difference between the samples of distributions. We quantify the sample quality of TurbDiff and the baselines by their Wasserstein-2 ($W_2$) distance to samples from the numerical solver. The $W_2$ distance between two sets of samples is the minimal average distance achievable for any matching between the two sample sets w.r.t. some underlying distance $d$. Formally, let $\mu, \nu$ be two empirical distributions over the sample sets we want to compare and $\Gamma$ be the space of all possible joint distributions that have $\mu$ and $\nu$ respectively as marginal distributions. Then the $W_2$ distance between $\mu$ and $\nu$ is defined as

$$W_2(\mu, \nu) = \left( \min_{\gamma \in \Gamma(\mu,\nu)} \mathbb{E}_{(x,y) \sim \gamma} d(x, y)^2 \right)^{\frac{1}{2}}. \tag{5}$$

In addition to being the basis of the ubiquitous Fréchet inception distance (FID) in generative image modeling (Heusel et al., 2017), $W_2$ is also used in video (Unterthiner et al., 2019) and molecule generation (Preuer et al., 2018). However, these metrics rely on learned embeddings from pre-trained models that are widely accepted to capture the important characteristics of individual samples.

It remains to specify the underlying distance function $d$. The Euclidean distance would be an obvious but lousy choice for a curious reason. Because of the chaotic nature and strong, small-scale fluctuations of turbulent flows, it is highly unlikely for two independent samples of a turbulent flow to be close under this distance. In fact, in our dataset, the smooth mean flow is closer to all samples than any two uncorrelated turbulent flow samples are to each other with respect to the Euclidean distance. This, however, would be antithetical to our goal to define a metric that rewards turbulent samples and discourages smoothing and mean flow samples.

**Turbulent kinetic energy (TKE)** Our first choice of distance compares samples by their TKE spectra. The TKE in each cell is defined as the quadratic velocity deviation from the mean velocity, $E_{\mathbf{i}} = \frac{1}{2} \| u_{\mathbf{i}} - \bar{u}_{\mathbf{i}} \|_2^2$. To get the spectrum, i.e. the spectral intensity as a function of the wavenumber, we take the 3D fast Fourier transform (FFT) of $E$ over a cuboid subset $C$ of the domain and integrate the squared frequency magnitude spherically over all 3D wavenumbers $\|(k_x, k_y, k_z)\| = k$:

$$E(k) = \int_{\mathbb{S}^2} \left| \int_{C \subset \Omega} E(\boldsymbol{x}) e^{-i(k\boldsymbol{v}) \cdot \boldsymbol{x}} \, \mathrm{d}\boldsymbol{x} \right|^2 \mathrm{d}\boldsymbol{v} \tag{6}$$

Then we define the distance between two samples as the $L_2$ distance between their log energy spectra,

$$d(\boldsymbol{X}^{(1)}, \boldsymbol{X}^{(2)}) = \| \log E_{\boldsymbol{X}^{(1)}} - \log E_{\boldsymbol{X}^{(2)}} \|_2. \tag{7}$$

We denote this distance by $W_{2,\mathrm{TKE}}$.

**Distributional distance** TKE spectra capture global patterns in the flow velocity by measuring the energy contained in the various spatial scales. However, the energy spectra neglect the pressure and are invariant to wrong positioning or orientations of flow patterns. To remedy this shortcoming, we introduce a second metric that measures if two samples contain similar velocity and pressure values in similar locations.

If we regard a turbulent flow as a stochastic process as is common, the velocities and pressures in each cell follow a marginal distribution $p(\boldsymbol{u}_{\mathbf{i}}, p_{\mathbf{i}})$. Then, we can call two samples close if their values in each cell follow similar marginal distributions. However, comparing cell values pointwise directly would make it highly unlikely for two samples from the same flow to be close, as explained above. Instead, we propose to group cells with similar marginal distributions together and interpret the values in each cell group as samples from the same marginal. Then we compare these sample sets by their $W_2$ distance and average over all cell groups.

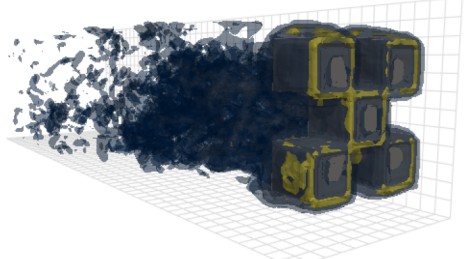
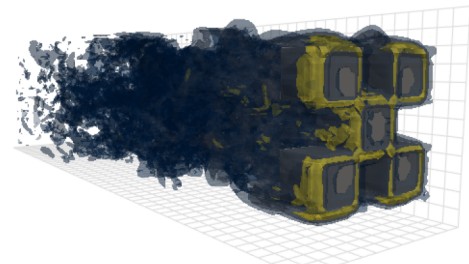

(a) Samples generated with TurbDiff
(b) Data sample for the same object

Figure 5: Example flows for a test object from our dataset showing isosurfaces of the vorticity $\boldsymbol{\omega}$.

Table 1: TurbDiff generates samples from scratch of similar quality in $W_{2,\mathrm{TKE}}$ & $W_{2,\mathcal{R}}$ distance to samples generated by the strongest regression baseline which takes a true data sample as initial state. In addition, our model is much faster to evaluate because it eschews the 10 minutes needed by the numerical solver to generate a realistic initial state for DilResNet-22. For models marked with (*), we have excluded runs that diverged due to error accumulation.

| | $W_{2,\mathrm{TKE}}$ | $W_{2,\mathcal{R}}$ | RMSE $\boldsymbol{x}_{\mathrm{max-TKE}}$ | Runtime [s] |
|---|---|---|---|---|
| TF-Net-init* | - | - | - | 1.834 |
| TF-Net-22* | 189 | 493 | 108 | 602 + 0.23 |
| DilResNet-init | 60 ± 47 | $4.6 \times 10^8$ | 42 ± 23 | 12.82 |
| DilResNet-22 | 2.15 ± 0.06 | 1.240 ± 0.001 | 2.5 ± 1.2 | 602 + 1.58 |
| TurbDiff (ours) | 3.9 ± 0.4 | 1.38 ± 0.04 | 5.9 ± 1.8 | 20.63 |

In particular, for each simulation, we approximate the marginal velocity distribution $p(\boldsymbol{u_i})$ of each cell with an isotropic Gaussian $\mathcal{N}(\boldsymbol{\mu_i}, \mathrm{diag}(\boldsymbol{\sigma_i^2}))$. Then, we cluster the cells into regions $\mathcal{R}$ with similar marginal distributions via k-means with the $W_2$ distance between isotropic Gaussians, which has a closed-form solution. Finally, we define our distance between two samples $\boldsymbol{X}^{(1)}$ and $\boldsymbol{X}^{(2)}$ as

$$d(\boldsymbol{X}^{(1)}, \boldsymbol{X}^{(2)}) = \left( \sum_{\mathcal{R}} \frac{|\mathcal{R}|}{\sum_{\mathcal{R}'} |\mathcal{R}'|} W_2^2 \left( \boldsymbol{v}_{\mathcal{R}}^{(1)}, \boldsymbol{v}_{\mathcal{R}}^{(1)} \right) \right)^{\frac{1}{2}} \tag{8}$$

where $\boldsymbol{v}_{\mathcal{R}}^{(j)} = \left\{ \left( \frac{\boldsymbol{u_i}^{(j)}}{\sigma_{\|\boldsymbol{u}\|}} \parallel \frac{\boldsymbol{\omega_i}^{(j)}}{\sigma_{\|\boldsymbol{\omega}\|}} \parallel \frac{p_{\mathbf{i}}^{(j)}}{\sigma_p} \right) \mid \mathbf{i} \in \mathcal{R} \right\}$ is the set of normalized velocity, vorticity and pressure at each cell in a homogeneous region. This metric includes the generated pressure as well as the estimated vorticity, which means that it also takes into account the velocities in neighboring cells. All components are normalized by their standard deviation over the training set to account for their different scales. We choose the $W_2$ distance with the 2-norm to compare the sample sets $\boldsymbol{v}_{\mathcal{R}}^{(j)}$ to emphasize modeling the extreme values which are important to represent the characteristic intermittency of turbulent flows, i.e. the heavy tails of their velocity and vorticity distributions (Jiménez, 2006).

In addition to velocity and pressure, we also include the estimated vorticity $\boldsymbol{\omega_i}$ because vorticity is a defining aspect of turbulence, as we have seen in Section 2. Furthermore, since we estimate the vorticity from the velocity via finite differences, including it in the distance makes it more discriminative because it includes distance from the multi-point marginal velocity distribution of cell $\mathbf{i}$ and its neighbors.

In our results, we denote this distance by $W_{2,\mathcal{R}}$. See Appendix E for more details on how exactly we compute each metric.

## 6.4 SAMPLE QUALITY

We train all models on the 27 training simulations in our dataset and then evaluate them on 9 unseen test simulations with unseen geometries. For each simulation, we generate 16 samples and evaluate our sample metrics against 16 true samples chosen equidistantly from the second half of the simulation. Choosing the data samples from the second half of each simulation ensures that they represent fully-developed turbulence, and picking them equidistantly means that the samples are completely uncorrelated. Figs. 5a and 5b show samples generated by TurbDiff and from the dataset for the same object. See Table 1 for the complete results.

First, we observe that our generative model generates samples very close to the data distribution in terms of $W_{2,\text{TKE}}$ and $W_{2,\mathcal{R}}$ distance. Only the DilResNet-22 baseline achieves better scores. However, this baseline needs a true sample from our target distribution $p_{\geq t_{\text{turb}}}(\boldsymbol{X} \mid \boldsymbol{T})$ generated via a numerical solver as initial state and then evolves it forward in time for 22 steps or 2.2 ms. As we trained the regression baselines on the smallest timestep in our data to allow them to resolve the dynamics of all scales accurately, see Section 2, and 22 roll-out steps still have tolerable error accumulation, the baseline is able to preserve the turbulent characteristics of its input state and achieve slightly better scores. If we evaluate the baselines in the more realistic setting denoted by the *-init* suffix without any support from a numerical solver and generate samples by unrolling from the pre-turbulent initial state of the numerical solver, the increased error accumulation of the longer roll-out decreases the sample quality immensely. However, this is the only setting that achieves the original goal of surrogate models to replace costly numerical solvers.

These results support our observation from Sections 3 and 4 that autoregressive models are severely challenged by the long roll-outs required to fully replace numerical solvers for 3D turbulence and that a generative approach sidesteps these difficulties elegantly.

**Runtime** Regarding runtime, TurbDiff outperforms the numerical solver by a factor of 30, as reported in Table 1. The solver needs roughly 10 minutes to transition from its initial state to fully-developed turbulence, whereas our model can generate a sample in about 20 seconds. While both baselines can be evaluated quicker if we consider just the model itself, the complete pipeline as a surrogate model requires the generation of a realistic initial state for the model, which takes 10 minutes to generate with a numerical solver. In the -init setting, the models avoid this additional runtime but suffer from significant error accumulation.

All model times represent the minimum achieved time on an NVIDIA A100, and we measured the solver time with 16-core parallelism on an Intel Xeon E5-2630.

**Estimating turbulence properties** In applications, samples from a generative turbulence simulator could be used to measure, for example, the effect of an object's shape on the turbulence patterns it causes, e.g. their size or energy. As an example, we estimate how far behind the object in our flow simulations the mean TKE takes its maximum, i.e. where the turbulent vortices caused by the object have built up the most kinetic energy. This location depends strongly on the shape of the object and requires accurate samples to evaluate. The root-mean-squared-error (RMSE) of the estimate from our samples reported in Table 1 is 5.9 cells or just 1.2 cm in a channel of 40 cm in length. This is significantly lower than the variability of the true location and provides an actual useable signal to practitioners compared to the baselines in the full-surrogate setting.

## 7 CONCLUSION

We have shown that generative modeling lets us access the full acceleration potential of neural surrogates for 3D turbulent flows by circumventing the need for long roll-outs or initial states from numerical solvers. Our dataset is an important building block for further research into surrogate models for engineering applications that challenges models to learn the turbulent dynamics of high-velocity flows and how objects influence the resulting vortex structures. Our new metrics for generative turbulence models measure succinctly if a model's samples follow the expected velocity and pressure distributions as well as if the turbulent kinetic energy is distributed correctly across spatial scales.

ACKNOWLEDGMENTS

This research was funded by the Bavarian State Ministry for Science and the Arts within the framework of the Geothermal Alliance Bavaria project.

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

## A  DATASET DETAILS

Our dataset consists of 45 simulations, each with a separate object in the flow that causes distinct turbulence patterns. See Fig. 6 for a visualization of all objects. Each simulation is in a channel of size 0.4 ×0.1 ×0.1 cm, which is discretized into 192×48×48 cells. The inflow velocity is 20 meters per second, and we simulate the flow for 0.5 s with a snapshot saved every 0.1 ms. This results in 5,000 flow states in total per simulation.

For the simulations, we use OpenFOAM, an industrial grade CFD solver, in LES mode. LES is an established method to produce realistic turbulent flow patterns in contrast to simpler Reynold's-averaged Navier-Stokes methods. It resolves the vortices above a cut-off scale and uses a simplified model to represent the effect of sub-grid scales so that the flow does not have to be spatially resolved all the way to the Kolmogorov scale, which is infeasible computationally as well as spatially in most situations and also for our dataset.

We choose a low viscosity of $\nu = 1 \times 10^{-5}$, which leads to an average Reynold's number of $\mathrm{Re} = \frac{\|u\|L}{\nu} = 2 \times 10^5$ and guarantees that fully-developed turbulence will occur.

Turbulent flow simulations need significant time to build up the turbulent flow state from the smooth initial state, i.e. initiate turbulent structures at all scales and break any symmetry that the simulation geometry has. The exact time depends on the boundary and initial conditions, i.e. also the object in the flow. For our dataset, this initial transition lasts for roughly 22 ms to 45 ms in simulation time, which corresponds to about 2,000 to 4,000 solver steps or 10 to 20 minutes of solver time on 16 CPU cores.

For training our generative model, we discard the first 25 ms so that it only learns the distribution of turbulent states and not the distribution of the initial transition phase. For regression models, we train on the full sequence to ensure they also learn the initial transition for the full roll-out experiments.

## B  DATA PREPROCESSING

OpenFOAM generates data within the simulation cells. However, for the baselines and our model, which rely on convolutions, we have to embed the data into 3D grids. To also represent the boundary, inlet, outlet, and inside of the geometries, we choose the following representation. The out-of-domain cells, e.g. in the objects, are represented with zeros. The outer boundary around the whole domain represents the Dirichlet boundary conditions and boundary layer types such as wall or inlet. During the construction of our grid-base representation, we keep track of the type of each cell to construct the cell type tensor $T$. See Fig. 7 for an illustration.

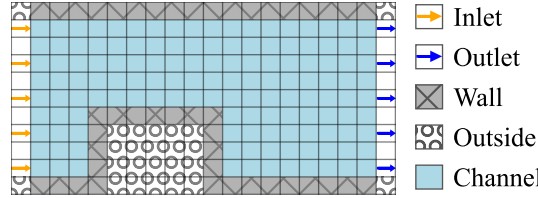

Figure 7: An illustrative 2D slice of our grid-base representation of the simulation data.

## C  BASELINE DETAILS

We implemented the two baselines TF-Netand DilResNetas described in the original papers and replaced all 2D convolutions with 3D convolutions. Further, to stay comparable to our model, we include the cell type information $T$ into the model by learning 8-dimensional embedding vectors $c(T)$ and concatenating them to the flow state $(X \parallel c(T))$. We choose all hyperparameters as in (Stachenfeld et al., 2022), i.e. DilResNet takes only the most recent step $X$ and predicts the delta to the next step $\hat{X} = X^{(t+1)} - X^{(t)}$ and uses 4 dilated convolutional blocks of 7 layers with residual connections. TF-Net takes the 6 most recent $X^{(t-5:t)}$ and predicts the next state $X^{(t+1)}$ directly.

## D  MODEL DETAILS

**DDPM**  The DDPM framework described in Section 4.3 has two hyperparameters, the number of steps $N$ and the noise schedule $\beta_n$ that controls how quickly the data samples are transformed into

Gaussian noise. We chose $N = 500$ and the log-linear signal-to-noise ratio (SNR) schedule from (Kingma et al., 2021) that scales $\beta_n$ such that the log-SNR of the data falls linearly from $1 \times 10^3$ to $1 \times 10^{-5}$. We have found that this schedule is especially suitable for turbulence data, because it leaves very little signal in the noisy training samples for large $N$. This counteracts the large extreme values in turbulence data and ensures that the distribution $\mathrm{q}(\boldsymbol{x}_N \mid \boldsymbol{x}_0)$ converges sufficiently closely to $\mathcal{N}(\boldsymbol{0}, \boldsymbol{I})$ to make the reverse process work well. See (Ho et al., 2020) for the full derivation of the DDPM framework and a definition of $\beta_n$.

**Architecture** We need to parametrize $\mathrm{p}_{\boldsymbol{\theta}}(\boldsymbol{x}_{n-1} \mid \boldsymbol{x}_n)$ such that it produces an estimate of $\mathrm{q}(\boldsymbol{x}_{n-1}|\boldsymbol{x}_0, \boldsymbol{x}_n)$. In particular, we use a U-Net with 4 levels of downsampling and apply a transformer at the coarsest scale. First, we encode the noise level $n$ into a 32 dimensional vector with a sinusoidal positional encoding (Vaswani et al., 2017) and then apply a 2-layer MLP to it. Then, each cell's velocity $\boldsymbol{u}$ and pressure $p$ features are concatenated with a 4-dimensional, learned cell type embedding and mapped linearly onto a 64 dimensional latent vector. Both, the latent feature vector tensor and the noise level embedding are then passed into a U-Net. The U-Net downsamples the spatial resolution 4 times by a factor of 2, applying a ResNet style block at each level (He et al., 2015). After the final downsampling step, we apply full attention between all downsampled cells. Finally, the resulting representation is upsampled again 4 times to the original resolution with an equivalent ResNet-style block applied at each level to the upsampled data concatenated with a skip connection from the downsampling. The final feature tensor is then mapped from the 64-dimensional latent space to the $(\boldsymbol{u}, p)$ data space via a final ResNet-style block.

**Training** We trained TurbDiff from 3 different seeds for 10 epochs with a batch size of 6. The optimizer is RAdam with a learning rate of $1 \times 10^{-4}$ and and exponential learning rate decay to $1 \times 10^{-6}$ over those epochs.

**Normalization** The data for all models is normalized with the maximum norm of $\boldsymbol{u}$ for the velocity and the maximum of $|p|$ for pressure. We choose to normalize by the maxima of each feature to handle the long tails of the pressure and velocity distributions, which show values of up to $10\sigma$ away from the mean. Lastly, for our diffusion model, we do not apply noise to the cell-type embeddings but concatenate them without noise at each diffusion step.

# E METRICS DETAILS

The TKE computes an FFT over the flow field. Since we integrate the spectral intensity over all spatial frequencies of the same magnitude, we would incur boundary effects if we take the FFT over a rectangular domain because higher frequencies could be computed along the long dimension of the domain than the shorter ones. To avoid this problem, we compute the the TKE distance separately over 48×48×48 cubes behind the object, in the middle of the flow and at the end of the channel and then compute the individual distances

$$d_{\mathrm{TKE}}(\boldsymbol{X}_a, \boldsymbol{X}_b) = \sqrt{\sum_{i=1}^{3} d_{\mathrm{TKE}}^2(\boldsymbol{X}_{a,\mathrm{block}\,i}, \boldsymbol{X}_{b,\mathrm{block}\,i})}. \tag{9}$$

We estimate the $L_2$ distance between two log TKE spectra $E_a(k)$ and $E_b(k)$

$$\|\log E_a - \log E_b\|_2^2 = \int_1^{23} (\log E_a(k) - \log E_b(k))^2 \, \mathrm{d}k \tag{10}$$

via Gauss-Legendre integration to get a highly accurate estimate. To evaluate $\log E(\boldsymbol{k})$ for non-integer $\boldsymbol{k} = (k_x, k_y, k_z)$, we interpolate the values linearly from the discrete FFT onto the points $\boldsymbol{k}$. This let's us get a highly accurate estimate of the total spectral intensity at a frequency $k$ estimating the spherical integral via the Lebedev integration method (Lebedev & Laikov, 1999).

For the distributional distance, we need to take care that no region of cells with approximately the same marginal distribution has too many cells, because we compute the exact $W_2$ distance within each region $\mathcal{R}$ and the runtime of $W_2$ scales with $O(n^3)$ where $n$ is the number of cells within each region. To avoid this problem, we split regions larger than 512 cells into groups of at most 512 cells

arbitrarily. We can split them arbitrarily because all the cells in the region behave in very similar ways anyway.

## F    LIMITATIONS

**Performance**    While we have shown that our approach reduces the time to first sample significantly compared to a CFD solver, a solution based on 3D grids will necessarily scale cubically in resolution per dimension. This could be alleviated by representing the data in different ways such as frequencies (Li et al., 2021) or more general function spaces (Lienen & Günnemann, 2022). Alternatively, hierarchical models such as the U-Net could be parallelized over multiple devices. Another aspect is that generative diffusion is known to produce high-quality samples but also for being expensive. This could be improved by faster sampling routines (Nichol & Dhariwal, 2021; Rombach et al., 2022; Song et al., 2021) or replacing the sampling routine altogether with a model (Biloš et al., 2021; Song et al., 2023).

**Equivariance**    Since TurbDiff is based on a basic 3D U-Net, it does not exploit any of the symmetries that turbulence as a physical process obeys, which could improve sample efficiency and generalization (Pope, 2000). This includes translation, rotation and reflection equivariance (Satorras et al., 2021; Finzi et al., 2020; Weiler et al., 2018; Gasteiger et al., 2021) but also the turbulence-specific Reynolds number similarity.

**Geometry**    With our grid embedding, we can represent any geometry given that the resolution is fine enough, however only to a 0th order approximation, i.e. as step functions due to the grid structure. Since sharp corners have a strong effect on turbulent flows, sample quality for smooth geometries would likely benefit from 1st order approximations with meshes (Lienen & Günnemann, 2022) or even smoother representations based on signed distance functions (Park et al., 2019). Mesh-based models could furthermore posses a certain invariance with respect to the mesh density as was demonstrated by (Lienen & Günnemann, 2022).

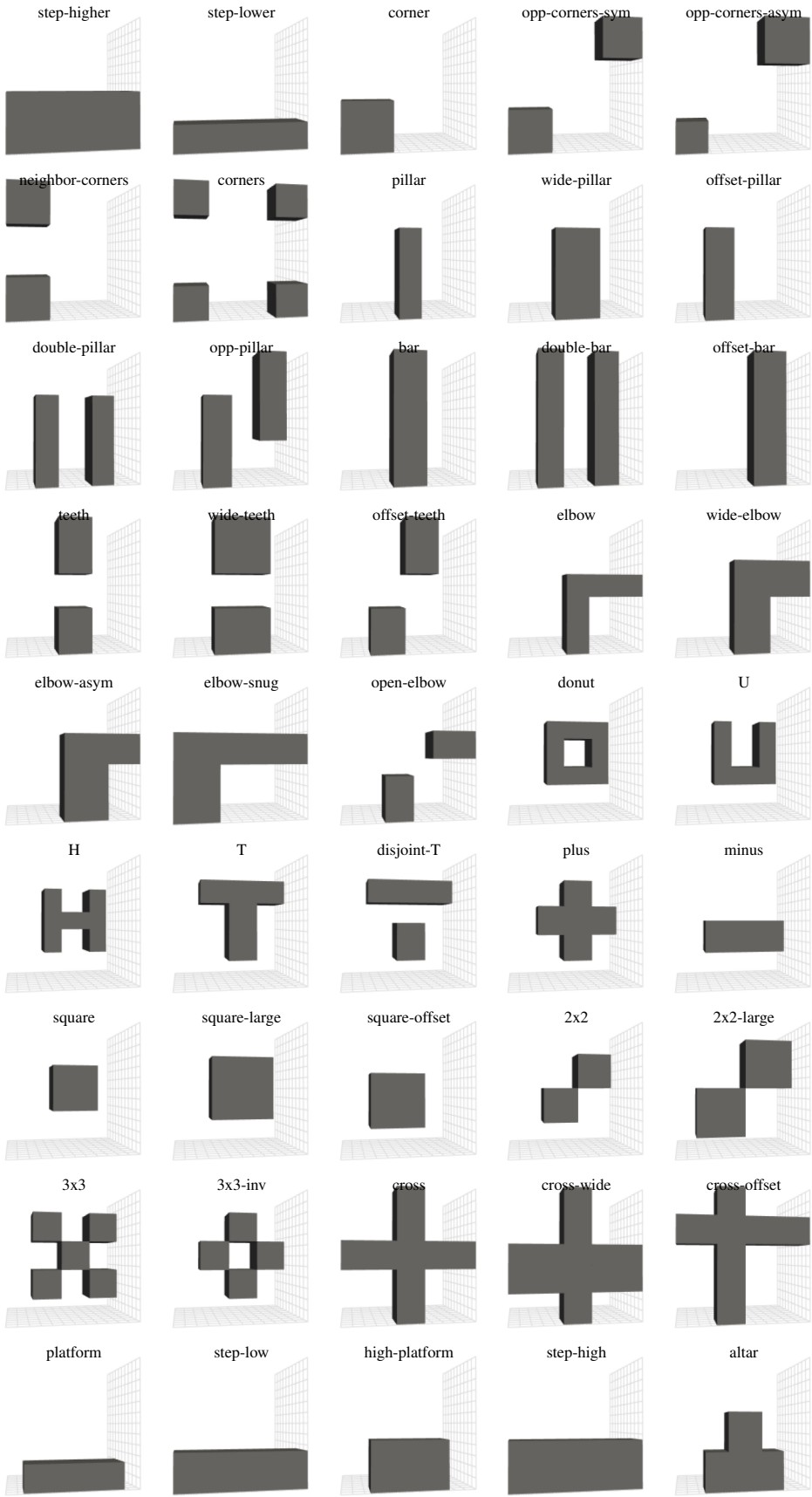

Figure 6: All objects in our dataset.

