# OpenReview forum: "From Zero to Turbulence: Generative Modeling for 3D Flow Simulation"
_ICLR.cc/2024/Conference — ICLR 2024 poster_

### Official Review · Reviewer_MYNP · 2023-10-26

**Soundness:** 3 good
**Presentation:** 3 good
**Contribution:** 3 good
**Rating:** 8
**Confidence:** 3

**Summary:**

The paper concerns simulation of 3D turbulent flows and replacing direct numerical solvers with autoregressive, learned models. The authors argue that turbulent flows are substantially harder to model properly in 3D than in the 2D case which many papers previously have targeted. They then propose a generative model for flow simulation that is independent of the initial state but still able to capture the distribution of fluid states with high-resolution in complicated settings with various objects affecting the flow.

**Strengths:**

- well-written and clearly presented paper
- important problem and method that can potentially have impact
- targets 3D instead of 2D and argues that flow simulations in this case is substantially harder
- works in challenging situations with boundary effects and objects influencing the flow

**Weaknesses:**

- while the paper presents a new method that is different from the generative flow methods in the literature, it is quite related to previous approaches and thus provide a somewhat incremental - but still important - contribution

Minor points:
- some parts need proofreading, e.g. "We conclude that many use cases can be solved by independent snapshots of possible flow states as well as by a classical numerical simulation. We conclude that many use cases can be solved by independent snapshots of possible flow states just as well as with a classical numerical simulation."

**Questions:**

- I think it is a little unclear that you write "Thus, we propose to circumvent the roll-out dilemma of autoregressive models by modeling the marginal distribution E_t[p((u, p)t | u0, p0)] directly, capturing the distribution of turbulent flows with a generative model.", but then in the next paragraph remove the dependence on the initial state and remove the expectation. As far as I can see, the stochastic process perspective has been used previously (e.g. ref Yang and Sommer), and what is done here is removing the condition on the initial state. I'm note sure where the expectation comes in.

---

> ### Author Response · Authors · 2023-11-20
>
> Thank you for your thorough review. We have uploaded a new version with improved wording.
>
> **Stochastic process perspective**
>
> While (Yang and Sommer, 2023) have applied a DDPM model to a 2D fluid prediction problem, they do not use or mention the stochastic nature of turbulent flows and present their approach in the usual time series forecasting framework, i.e., predicting a future state given a current state. In contrast, we specifically exploit the insight that 3D turbulent flows behave chaotically with many small-scale and short-lived features to propose generative modeling as an alternative form of neural surrogate model for these systems. Furthermore, our dataset goes beyond their data as well as many other datasets in the machine learning community. First, by simulating in 3D, our data exhibits more small-scale behavior than a comparable 2D dataset. Second, the objects in the flow and how they influence the turbulence patterns make our dataset and paper particularly relevant for practical applications.
>
> **Q: Why do you introduce the initial state $\boldsymbol{u}\_{0}, p\_{0}$?**
>
> We have changed the term $\mathbb{E}\_{t}[\mathrm{p}((\boldsymbol{u}, p)\_{t} \mid \boldsymbol{u}\_{0}, p\_{0})]$ in Section 4 to $\mathbb{E}\_{t}[\mathrm{p}((\boldsymbol{u}, p)\_{t} \mid \boldsymbol{u}\_{0}, p\_{0}, t)] = \mathrm{p}((\boldsymbol{u}, p) \mid \boldsymbol{u}\_{0}, p\_{0})$ to clarify that our model learns to sample from the flow distribution marginalized across time, i.e., approximates the underlying distribution with a stationary distribution.
>
> In Section 4.1, we propose the generative turbulence simulation task by noting that the above marginal distribution $\mathrm{p}((\boldsymbol{u}, p) \mid \boldsymbol{u}\_{0}, p\_{0})$ is independent of the choice of initial state $\boldsymbol{u}\_{0}, p\_{0}$ under the assumption of ergodicity. It may seem unnecessary to introduce the initial state given the ergodicity assumption, however, this is important to convey our method clearly to the majority of readers who are likely most familiar with neural surrogate models in terms of time series modeling.

---

> > ### Comment · Reviewer_MYNP · 2023-11-22
> > **Response**
> >
> > Thank you for the response. The response has not changed my rating.

---

### Official Review · Reviewer_JYNa · 2023-11-01

**Soundness:** 3 good
**Presentation:** 3 good
**Contribution:** 3 good
**Rating:** 8
**Confidence:** 4

**Summary:**

Methods
- The paper focuses on the topic of 3D turbulence generation, by modeling it as a generative task to overcome the roll-out dilemma of autoregressive models.
- The paper derives an appropriate generative diffusion model and the discretization on 3d domain for the 3D turbulence generation. Dirichlet conditions are handled in the model. Cell types are handled via learned cell type embeddings.
- The network is based on denoising diffusion probabilistic models (DDPMs). The network architecture employed in the diffusion model extends the U-Net framework into the 3D domain, incorporates a transformer (after 4 level of down-sampling) to enable global communication across the simulation domain.

Experiments
- The paper provides a new 3D turbulence dataset, consisting of 45 simulations of an incompressible flow in 192 × 48 × 48 grid cells.
- 9 out of 45 samples from the dataset are used to validate the method. From the results, the proposed method is able to generate realistic 3D-flow samples that can capture the turbulent flows caused by unseen objects.
- The paper proposes metrics for evaluating the quality of the generated flows by utilizing properties of fluids, including turbulent kinetic energy, marginal velocity, vorticity and pressure distribution, and the location of the strongest turbulent vortices.
- The method is compared with two autoregressive models (Turbulent Flow Net and DilResNet) and each with two different settings to convert them to generative models, showing good performance compared with them.

**Strengths:**

- This paper leverages generative diffusion models, which are more commonly associated with image and video tasks, for turbulence simulations, providing a new approach for 3D turbulence generation.
- In the experiment section, the metrics are well designed to capture the characteristics of fluids, and the proposed method shows good performance in terms of both accuracy and runtime.
- The paper gives a good discussion regarding the challenges of employing autoregressive models for 3D turbulence generation.
- In addition to the method, the paper introduces a new 3D turbulence dataset that can benefit further research in the field of turbulence generation.

**Weaknesses:**

- It would be good (but not necessary in this paper) to see how the method performs on learning and generating more complex fluid phenomena, such as fluid interactions with air, and generating classical fluid phenomena like the Karman vortex street.

**Questions:**

- In Fig5 and Fig6, what’s the value that is visualized? the norm of velocity? pressure?
- Would it help to pre-processing the data, e.g. rotations, flips to augment the dataset?
- Is it relatively easy or hard to apply this method to more complex turbulent generation, e.g., with fluid–solid interaction?
- The grid size used for generating dataset is 192 × 48 × 48. In fluid simulations, larger grid size are often used to achieve higher accuracy. Is this grid size chosen due to performance limitation of the CFD solver?

---

> ### Author Response · Authors · 2023-11-20
>
> Thank you for your valuable review.
>
> **Q: Which quantity is visualized in Figures 5 and 6?**
>
> Figures 5 and 6 show isosurfaces of the norm of the vorticity $\omega$ as in Figure 4, which visualizes the local rotationality of the velocity field, an essential aspect of turbulence. We have updated the paper accordingly.
>
> **Q: Would data augmentation improve performance?**
>
> Data augmentation could certainly be helpful since the underlying physics also obeys various symmetries. Ultimately, we believe that the most promising direction for future research is equivariant generative models that obey the same symmetries as the Navier-Stokes equation, e.g. translation and rotation equivariance but also uniform motion and scaling equivariance.
>
> **Q: Can your approach be applied to more complex scenarios?**
>
> In principle, it is possible to encode, for example, phase information to model multiphase flows or use a graph neural network as the denoising model to work with an arbitrary mesh discretization. In this work, we chose single-phase flows with a grid discretization to focus on the question whether generative models are viable neural surrogate solvers isolated from other, orthogonal modeling aspects, such as multi-scale graph neural networks.
>
> **Q: Why did you choose 192×48×48 as your grid resolution?**
>
> Our choice of resolution balances dataset size, solver performance, and model memory requirements against the small-scale turbulence phenomena important to our work. We picked a small time step of $\Delta t = 0.1\mathrm{ms}$ between snapshots to ensure that it is possible for the autoregressive baselines to learn the short-lived dynamics of the smallest scales in the turbulent flows. However, these choices already result in a dataset of 9 TB. Even a moderate increase in resolution to 256×64×64 would more than double the size and strain our available computing resources. Furthermore, computational requirements grow quickly with resolution for operations like convolutions and attention. The resolution we chose already limited the batch size in training, even though we trained on A100s and chose a U-Net backbone for our model to minimize memory requirements. Even more efficient architectures are an important research area for future applications in 3D simulations.

---

### Official Review · Reviewer_oPBx · 2023-11-02

**Soundness:** 3 good
**Presentation:** 4 excellent
**Contribution:** 3 good
**Rating:** 5
**Confidence:** 3

**Summary:**

This work proposes a generative model based on a diffusion process for one-shot generation of solutions to turbolent flows.
The approach is remarkable in his simplicity and potential applicability.

**Strengths:**

The approach of using a one-shot generation for solutions to turbolent flow is quite remarkable, as well as the fact that results seem to be rather impressive on such a small data regime.
The experiments show training on 27 airfoils, 9 for val and 9 for testing. It is indeed surprising the model can generalize in such a small regime, perhaps indicative of the fact that the model can pool information across all cells/volumes, thus perhaps capturing somehow the intrinsic physics of the problem.

**Weaknesses:**

There is no mention of releasing the data-set and the source-code. This is a major issue for reproducibility.
The size of the data-set is also surprisingly small. While it's quite remarkable the model seem to capture the essence of the problem is such a small data-regime, I am also quite puzzled by that.
It would be great to scale to deep-learning type of sample size (10^5 samples - of course what's a sample in the CFD case is a bit up to interpretation).

I also agree the limitations of A#F are palpable: especially the lack of equivariance and the geometry limitations are quite important.

Of course this also means the architecture has much to grow, once such limitations are addressed, but this also points to the fact this is likely an initial work, and possibly not ready yet for primetime publishing in top venue like iClear.

Finally, the speedup is somewhat disappointing: the nominal 30x achieved is really not so important when considering that A100 vs Xeon-2630 is probably already 50x faster (depending on workloads).

**Questions:**

Q1: plans to release code and database
Q2: plans to scale up experiments
Q3: why does it generalize in such a small data regime? any thoughts?

---

> ### Author Response · Authors · 2023-11-20
>
> Thank you for your thoughtful review.
>
> **Source code and data**
>
> Please find the link to the anonymized source code and a subset of the data in our non-public global reply.
>
> We will release the source code publicly upon acceptance of the paper. Regarding the dataset, we are still looking for a viable hosting solution that can deal with the size of our dataset (9 TB). Alternatively, the accompanying scripts make it easy to generate the data yourself.
>
> **Dataset size**
>
> Our dataset consists of $27 + 9 + 9 = 45$ simulations for training, validation, and testing, respectively, with each simulation comprising 5000 snapshots, one taken every 0.1 milliseconds for 0.5 seconds. This gives us $27 \times 5000 \approx 10^5$ samples for training minus the ones corresponding to the transition phase from the initial condition to turbulence. The raw data takes up roughly 150GB for each simulation with a further 50GB after preprocessing the data into an HDF5 file. Overall, this puts our dataset at 9TB for all 45 simulations or 2.25TB for the preprocessed data only. We have amended the paper to clarify this in the main text.
>
> Given the significant size of the dataset, further scaling up our experiments is not viable given our hardware constraints.
>
> **Generalization**
>
> Our training set comprises over $10^5$ samples based on 27 objects representing a large collection of turbulence patterns. The objects are sufficiently varied that the interactions between different components, e.g. corner configurations or shape widths, and their effects on downstream turbulence can be learned and transferred to unseen objects.
>
> **Equivariance and mesh geometries**
>
> While we are aware of the limitations concerning equivariance and mesh geometries (described in Appendix F), we respectfully disagree with the reviewer that these are fundamental weaknesses of our approach. As our main contribution is to establish generative models as viable neural surrogate solvers for turbulent flows, we opted for the U-Net architecture widely used for 2D and 3D data. However, our approach is not limited to it; any architecture, including equivariant or graph-based ones, can be plugged in as a backbone. We hope that our framework will benefit from future model developments in these areas.
>
> **Speed-up**
>
> Though GPUs can outperform CPUs in many tasks, it does not mean that CFD solvers benefit from GPUs in the same way. In fact, [OpenFOAM has been reported to achieve a performance boost of just 2x on GPUs](https://www.esi-group.com/sites/default/files/resource/other/1806/8th_OpenFOAM_Conference_Cineca_Spisso_2_0.pdf). Accordingly, a speed-up of 30x is highly relevant in practice regardless of the hardware differences. Further, please note that the performance of our model is constant with respect to the simulation parameters. In contrast, the runtime of the numerical solver depends on the velocity field $\boldsymbol{u}$ because it needs to scale its internal time step $\Delta t$ to keep the simulation stable.

---

### Official Review · Reviewer_DVzo · 2023-11-08

**Soundness:** 3 good
**Presentation:** 4 excellent
**Contribution:** 3 good
**Rating:** 6
**Confidence:** 4

**Summary:**

The paper proposes a novel modeling method of turbulence flow based on generative models. It consists in learning the distribution of turbulent flow states without the need to be tied to the initial flow state. The proposed model is augmented with Dirichlet boundary conditions to learn physically meaningful turbulent flow representations.

**Strengths:**

1/ Originality and significance:
Turbulence modelling is a crucial task in many engineering applications. However, the problem is still open and challenging. In the litterature several methods have been proposed to tackle the problem from different points of views but the proposed solutions are either expensive/ intractable in practice or are very complex to set. In this work, the originality of the work is in its simplicity of usability while providing good turbulent flow results in a raisonnable time. Hence, the proposed model is scalable.
Moreover, it is based on a generative model (diffusion model  DDPM) that has achieved significant leaps in a wide spectrum of applications. Augmenting DDPM with physics constraints like Dirichlet enables generating 3D turbulent flows of good quality. The strength of the proposed model is related to the fact it learns a manifold of turbulent flow states without being tied to the initial flow state.

2/ Quality:
The methodology of the proposed work is rigorous. A targeted 3D turbulent flow dataset is generated specifically to assess the capabilities of the proposed generative turbulent flow model in the appropriate conditions. Moreover, data and task dependent metrics are introduced to evaluate to the model in physical way.

3/ Clarity:
The paper is very clear and simple to follow. It is well structured.

**Weaknesses:**

The proposed work lacks of ablation studies and baselines both quantitative and qualitative analysis. Diving into the different blocks of the architecture to understand their impact/contribution is of great importance and can give more insights. Especially that the proposed model is a combination of sophisticated blocks: DDPM, U-NET, transformers. One would like to understand how these blocks cohabitate to generate good quality of turbulent flow.

Figure and the details on the architecture are not sufficient.

Invariance to meshing / discretization scheme has not been discussed.

**Questions:**

1/ Have you tested the proposed model on unstructured data like graph-meshes and cloud of points ? to what extent, it is applicable ? turbulent flow are irregular, regular grids can hide some patterns

2/ What are the capabilities of the proposed model at the boundary layers ? do you have any metrics on the surface of the geometry ? (most challenging part)

**Details Of Ethics Concerns:**

No concern.

---

> ### Author Response · Authors · 2023-11-22
>
> Thank you for your careful review.
>
> **Model architecture**
> As our main aim is to investigate whether generative models are viable neural surrogate solvers for 3D turbulent flows, we have purposefully chosen each component of our model. DDPM is an established, powerful framework for generative modeling in the image domain, which we chose as the basis of our generative model. U-Nets are well known as a capable neural architecture for grid-based data that produce outputs of the same dimension as the inputs while also creating a compressed internal representation through first down- and then up-scaling combined with skip connections. The spatial downscaling is important for our model so that we can apply a transformer at the coarsest scale and facilitate global communication between all parts of the domain without being held back by the quadratic scaling of attention. Consequently, the three components work together as one unit and neither can be meaningfully ablated.
>
> To emphasize the interplay between the components, we have expanded the description of the model in Appendix D. Furthermore, please see our global reply for the project's complete source code.
>
> **Domain representation**
> To focus on our primary research objective of investigating generative modeling for 3D turbulent flows, we need to disentangle it from other complex issues, such as the representational capacity of graph neural networks. Therefore, from the beginning, we decided to work with a regular grid discretization, which has been thoroughly researched in the ML community, particulary in the image domain, with established components like CNNs and U-Nets. As such, we did not test our model on irregular discretizations such as meshes or point clouds.
>
> **Invariance to discretization scheme**
> Since our model is based on convolutional layers through the U-Net and uses a transformer for global communication between cells, it is fully convolutional and, therefore, in principle, able to transfer to regular grids of different sizes. However, as convolutional layers do not use the physical distance information between cells, our model is not transferable to finer or coarser grids than it was trained on, i.e., the grid's physical resolution needs to match the training data. We have added a sentence discussing invariance with respect to discretization schemes to Appendix F.
>
> **Boundary layer**
> In this work, we focus on modeling the downstream turbulence patterns away from the boundary and the object in the flow influences them. Accurately resolving the boundary layer on a regular grid would require a much higher resolution than is feasible given our storage requirements, as our dataset at its current resolution already needs over 9 TB of disk space.

---

> > ### Comment · Reviewer_DVzo · 2023-11-30
> > **Response to the authors**
> >
> > I thank the authors for providing detailed explanations and for the exhaustive work.  l keep my original score.

---

### Author Response · Authors · 2023-11-20

We would like to thank the reviewers for their constructive feedback.

Please find the anonymized source code at the following link: https://figshare.com/s/e4dc4dbd6604d9d60b46

Besides code for the model, metrics, and preprocessing, the download also contains scripts to generate the complete dataset from scratch. Additionally, we have added 0.05 seconds (500 frames) of one of the simulations for testing purposes.

---

### Meta-Review · Area_Chair_WxKm · 2023-12-11

**Metareview:**

The paper creates a 3D temporal LES dataset of turbulent flows with obstructing objects of different geometries using openfoam. It trains a standard denoising diffusion model on a regular grid representation of the flow. The setup of the flow cases, data and its representation, and the DGM method are basic. The reviewers have generally found the setup of the work as a whole original, the results significant, and the presentation well done. The AC agrees and despite the simplicity of the aspects, finds it original within the ML community which can promote more advanced work in the field. Furthermore, the results are, in fact, promising compared to the more conventional autoregressive approaches in ML. The AC, thus, recommends acceptance.

**Justification For Why Not Higher Score:**

The paper's setup is quite basic. It is certainly worth being spread since it has a novel setup with significant results but does not particularly stand out in the pool of acceptable papers in any ML or application aspect.

**Justification For Why Not Lower Score:**

The paper has clear contributions that can lead a new direction for an important problem fundamental to several natural science applications. There is no fatal concern. It should be accepted.

---

### Decision · Program_Chairs · 2024-01-16

Accept (poster)